# Impact of COVID-19 on Lifestyle Habits and Emotional State of Childhood Cancer Survivors and Their Parents

**DOI:** 10.3390/nu15183932

**Published:** 2023-09-11

**Authors:** Maya Yardeni, Omer Zvi Shaked, Reut Pienik, Michal Ben-Ami, Eve Stern, Hana Golan, Dalia Waldman, Doron Gothelf, Dalit Modan-Moses

**Affiliations:** 1Division of Pediatric Hemato-Oncology, The Edmond and Lily Safra Children’s Hospital, Sheba Medical Center, Tel Hashomer 5262000, Israel; hana.golan@sheba.health.gov.il (H.G.);; 2Pediatric Endocrinology and Diabetes Unit, The Edmond and Lily Safra Children’s Hospital, Sheba Medical Center, Tel Hashomer 5262000, Israel; omerskd@gmail.com (O.Z.S.); chavistern@doctors.org.uk (E.S.); 3Faculty of Medicine, Tel Aviv University, Tel Aviv 6997801, Israel; doron.gothelf@sheba.health.gov.il; 4The Child Psychiatry Division, The Edmond and Lily Safra Children’s Hospital, Sheba Medical Center, Tel Hashomer 5262000, Israel; 5Sagol School of Neuroscience, Tel Aviv University, Tel Aviv 6997801, Israel

**Keywords:** childhood cancer survivors, COVID-19, lifestyle habits, anxiety, depression

## Abstract

Objective: The COVID-19 pandemic and associated social distancing measures affected the physical and emotional state of children and parents worldwide. Survivors of childhood cancer may be particularly vulnerable to these effects. We aimed to evaluate the lifestyle habits and emotional states of childhood cancer survivors and their parents during the COVID-19 outbreak. Methods: Lifestyle habits and emotional distress were assessed in 43 childhood cancer survivors (aged 8–21 years) and their parents before and during the COVID-19 lockdown, using the PROMIS anxiety and depression modules and the “Mabat Youth” questionnaire. Results: Most parents (80.5%) reported eating more family meals during home confinement compared to their usual routine. Patients’ physical activity levels did not change significantly during confinement, leisure-related screen time nearly doubled (*p* < 0.001), and sleep duration increased (*p* = 0.006). Anxiety levels of children (*p* = 0.045) and parents (*p* = 0.02) increased during confinement compared to pre-pandemic levels, with no significant changes in depression levels. Conclusions: Contrary to concerns regarding lifestyle habits during the COVID-19 lockdown, eating behaviors of childhood cancer survivors improved, sleep duration increased, and physical activity remained unchanged. Still, screen time increased significantly. Parents of childhood cancer survivors reported higher anxiety levels for themselves and their children during home confinement. Our findings may assist medical and psycho-social teams in guiding parents of cancer survivors during similar circumstances in the future.

## 1. Introduction

The COVID-19 pandemic and the restrictions aimed to mitigate its spread affected the physical and emotional states of children and parents worldwide. Following publications from China during the early stages of the COVID-19 outbreak [1], multiple studies showed high anxiety and depression rates in adults [2,3,4,5,6,7,8] as well as in children and adolescents [9,10,11,12,13]. Female gender, age, and lack of exercise were identified as risk factors for depression and anxiety [1,3,5,6,12,13], as was living in highly epidemic areas [9,11,13].

Self-isolation could also adversely affect lifestyle habits. Access to fresh foods may have been limited, while stress and boredom could result in dysfunctional eating behaviors. Moreover, containment measures limited opportunities for physical activity and may have promoted sedentary behaviors [14]. Furthermore, the interaction between lifestyle changes and stress caused by confinement could aggravate detrimental effects on children’s mental health [10]. Results of studies assessing the eating habits and physical activity of adults during the COVID-19 lockdown were inconsistent [10,14,15,16]. Regarding children and adolescents, several studies showed higher consumption of nutritious food and reduced consumption of fast food during lockdown [17,18,19,20]; however, there was a significant increase in screen time [20,21,22,23,24]. Results regarding physical activity were inconsistent, with most studies showing a decrease in total physical activity [18,21,22,23,24,25], while others reported no change in time devoted to physical activity [20]. 

Several studies showed a significant association between a history of prior physical illness and the psychological impact of the COVID-19 pandemic [1,6]. Concerns included being severely affected if infected, and reduced quality of healthcare during the pandemic [26]. Hence, children and adolescent cancer survivors could be at increased risk for adverse psychosocial outcomes during this period. It has been shown that depression and anxiety rates are elevated among childhood cancer survivors (CCS) and their parents [27]. The risk and course of COVID-19 in CCS have not been well characterized; however, it has been suggested that children with cancer have increased COVID-19-associated morbidity and mortality [28]. Furthermore, late complications of cancer therapy, such as heart failure, chronic lung disease, and impaired immunity, are considered risk factors for a severe course of COVID-19. Accordingly, the International Late Effects of Childhood Cancer Guideline Harmonization Group issued a consensus statement recommending additional precautionary measures to reduce the risk of COVID-19 exposure/infection in the workplace or home, seeking medical advice early, and alerting healthcare providers about survivors’ medical history [29]. This could cause confusion and fear among survivors and their parents. Indeed, in a study by Prasad et al., nearly a third (29.5%) of CCS had new-onset health concerns that developed over the first six months of the pandemic, and 36% of survivors reported concerns about contracting COVID-19 infection and how it would affect their health [30]. Moreover, the long period of quarantine could evoke traumatic memories from periods of isolation during the medical treatment of cancer [31].

In the current study, we investigated the impact of COVID-19-associated social distancing measures on lifestyle habits and emotional states in a cohort of CCS and their parents. We assessed eating behaviors, physical activity, and sleep habits among survivors and parents during the lockdown period and compared them to levels documented prior to the pandemic. In addition, we compared levels of anxiety and depression among survivors and parents during the lockdown period to prior data. 

## 2. Materials and Methods

Participants were recruited from a cohort comprising parent–child dyads who took part in a larger-scale study investigating chronic stress and lifestyle habits in CCS and their parents. Inclusion criteria were patients aged 8 to 21 years, at least one year after completion of treatment, with both the child and parent capable of completing self-assessment scales. Fifty-six consecutive parent–child dyads answered baseline questionnaires between 1 July 2019 and 18 February 2020 (see timeline of COVID-19 in Israel, Figure 1). For the current study, 54 of these parents were contacted and invited to answer an online questionnaire (one mother was excluded as her daughter had a relapse of her disease shortly after responding to the baseline questionnaire, and another one was excluded due to her own medical circumstances). Forty-three parents responded to the “Lockdown” questionnaires between 7 and 26 April 2020. At that time, children in Israel had been out of school for over four weeks, and strict restrictions had been in place for more than three weeks. Restrictions were gradually eased starting 19 April, and on 26 April, stores and many services were opened (Figure 1). Hence, parents who did not respond to the questionnaires by that time were excluded. These eleven non-participants included two who did not respond to recruitment phone calls and nine parents who agreed to participate but did not fill out the questionnaire. There were no active refusals. 

The study was approved by the Institutional Review Board (5312-18-SMC). Written informed consent was obtained from all parents prior to answering the baseline questionnaires; oral consent was obtained for participation in the current study during the recruitment call, with the approval of the Institutional Review Board.

### 2.1. Assessment

Symptoms of depression and anxiety in patients and their parents were assessed using the PROMIS (Patient-Reported Outcomes Measurement Information System) Adult, Pediatric, and Parent-proxy 1.1 Short Forms. The questionnaires have been translated into Hebrew and validated as described before [32,33]; each questionnaire consists of 6–8 items. At the time of baseline evaluation, the patients filled out the pediatric depression and anxiety questionnaires, while the parents filled out the self-report depression and anxiety questionnaires as well as the parent-proxy questionnaires. During the lockdown period, the parents filled out the self-report depression and anxiety questionnaires and the parent-proxy questionnaires, while their children were not invited to answer questionnaires. For the pediatric and parent-proxy PROMIS questionnaires, a T-score higher than 55 was considered indicative of moderate to severe symptoms of anxiety and depression, and for the parent self-report questionnaires, a T-score higher than 60 was considered indicative [34].

Eating habits, physical activity, and sedentary behavior were assessed using the “Mabat Youth” questionnaire. The questionnaire has been used in a national survey and validated in healthy Israeli children and adolescents [35]. The full questionnaire was administered at the baseline evaluation and filled out by the patients; for the current study, a shortened version was used, and it was administered to the parents. 

Study questionnaires were digitalized and sent to participants, and data were collected and managed using REDCap electronic data capture tools hosted at the Sheba medical center [36,37]. 

### 2.2. Clinical Data

Pertinent demographic and clinical data were obtained from the patients’ medical charts. Children’s height and weight were measured during the baseline visit. Body mass index (BMI) was calculated based on the formula weight (kg)/height (m)^2^. Height and BMI standard deviation scores (SDS) were calculated using age- and sex-specific growth data (based on the Centers for Disease Control and Prevention’s Year 2000 Growth Charts).

### 2.3. Statistical Methods

Paired samples comparisons were performed using the paired *t*-test. Log transformation was used in order to achieve the normal distribution of total times of physical activity, household chores, book reading, and playing musical instruments. The proportions of overweight participants in relation to the normal population were compared using the χ^2^ test. The McNemar Test for paired data was used to assess changes in eating behaviors between the two time points. Pearson correlation coefficients were calculated to assess the relationship between PROMIS-derived anxiety and depression scores, the duration of physical and sedentary activities, and sleep duration. Two-sided *p* < 0.05 was considered significant. Calculations were performed using SPSS software, version 25.0 (IBM, Armonk, NY, USA). 

## 3. Results

### 3.1. Baseline Characteristics of Patients and Parents

Forty-three parents (aged 45.7 ± 5.6 years, mothers = 31, fathers = 12) participated in the current study. The patients (males = 24, females = 19) were 14.1 ± 3.3 years old (range 8.4–20.5 years) at the time they filled out the baseline questionnaires, and were 6.2 ± 3.7 years (range 1.11–14.4 years) after completion of their anti-cancer treatment. Fourteen patients had recovered from leukemia, five patients from Lymphoma, eighteen patients from brain tumors, and six patients from other solid tumors. 

Patients’ mean height SDS (standard deviation score) at the time of baseline questionnaires was −0.49 ± 1.20, and mean BMI SDS was 0.27 ± 1.16 (range −2.36–1.96), with 32.6% of the patients being overweight or obese (BMI ≥ 85th percentile) [38]. This rate was not significantly different compared to recent data from Israel [35].

### 3.2. Eating Habits

On the baseline questionnaires, 24.3% of the children reported eating their main meal while screen-watching more than three times a week, compared to 21.4% during the lockdown period (*p* = 1.0). Still, 21.4% of the parents reported that their child had meals in front of a screen more frequently during the lockdown period, while 19% reported that their child had meals in front of a screen less frequently compared to the pre-lockdown period, and 59.5% reported no change.

Eighty percent of the parents (n = 34) reported having more family meals during the lockdown period compared to their usual routine, while 19.5% reported no change in this respect (no parent reported having fewer family meals). Accordingly, at the time of baseline evaluation, 26.8% of the children reported eating their main meal alone more than three times in a week, while during the lockdown period, only three children (7.5%) had their main meal alone more than three times in a week (*p* = 0.057). 

Twenty percent of the families reported ordering take-out food during the lockdown period. Only two parents (5.1%) reported ordering more take-out food during this period, twenty-two parents (56.4%) reported ordering less frequently compared to the pre-lockdown period, and fifteen parents (38.5%) reported no change. 

### 3.3. Physical Activity 

Table 1 depicts physical and sedentary activity at baseline and during the lockdown. At the time of baseline evaluation, 90.7% percent of the participants reported routinely performing physical activity, with a mean of 334.7 ± 356.5 min/week (range 25–1530). During the lockdown period, the reported rate of physical activity was similar (89.7%). The duration of activity was lower during the lockdown period, but not significantly so (247.5 ± 359.1 min/week, range 15–2100, *p* = 0.115) (Table 1 and Figure 2). Types of activity are shown in Table 2. As could be expected, during the lockdown period, patients engaged less in outdoor activities, but they did find alternative indoor activities. 

Leisure-related screen time nearly doubled during the lockdown period (947.3 ± 572.8 min/week vs. 477.0 ± 352.9 min/week, *p* < 0.001). Six children (14%) spent time in front of screens only during the lockdown period. Time devoted to reading books also increased significantly (*p* = 0.022). Time spent helping with household chores and playing musical instruments increased, but not significantly so (Table 1 and Figure 2). 

Total physical activity at baseline was highly correlated with total physical activity during the lockdown period (r = 0.617, *p* < 0.001). Similarly, total screen time at baseline was significantly correlated with total screen time during the lockdown period (r = 0.61, *p* = 0.001). There was no correlation between total physical activity and screen time at any of the time points. 

### 3.4. Sleep

According to the baseline questionnaires, on average, children went to bed at 10 p.m. during school days (range: 7:30 p.m.–1:00 a.m.) and at 11:42 p.m. during school holidays (range: 8:30 p.m.–3 a.m.). During the lockdown period, the average hour of going to bed was similar to school holidays—11:55 p.m. (range: 8:30 p.m.–3 a.m.). 

Sleep duration during school days was 8.3 ± 1.3 h, significantly (*p* = 0.006) shorter compared to sleep duration during the lockdown period, which was 9.2 ± 1.7 h (Figure 2). Sleep duration was inversely correlated with children’s age during school days (r = 0.480, *p* = 0.002) but not during the lockdown period (r = 0.039, *p* = 0.804). 

### 3.5. Anxiety and Depression

Figure 3 depicts the PROMIS depression and anxiety scores of the study participants. At the time of baseline evaluation, the mean self-reported PROMIS depression and anxiety scores of the patients were 47.54 ± 8.26 and 47.56 ± 9.40, respectively. Thirteen patients (30.2%) had PROMIS scores consistent with moderate or severe symptoms of emotional distress. Of these, four had scores consistent with depression, three had scores consistent with anxiety, and six had scores consistent with both depression and anxiety. 

According to the parents’ report (i.e., the parent-proxy questionnaires), the mean PROMIS depression score at baseline was 47.96 ± 8.52, and the anxiety score was 47.35 ± 9.69. There was no significant difference between the children’s self-report scores and the parents’ proxy report, either for depression (*p* = 0.766) or for anxiety (*p* = 1.0). 

During the lockdown period, parent-proxy anxiety scores increased significantly (50.26 ± 12.37, *p* = 0.045), but there was no significant change in the parent-proxy depression scores (49.55 ± 9.00, *p* = 0.279). 

Parent-proxy anxiety scores were significantly correlated with parent-proxy depression scores at both time points (baseline: r = 0.481, *p* = 0.001; lockdown: r = 0.579, *p* < 0.001). 

At the time of baseline evaluation, the mean self-reported PROMIS depression and anxiety scores of the parents were 46.33 ± 6.49 and 51.42 ± 7.22, respectively. During the lockdown period, parents’ anxiety scores increased significantly (53.84 ± 6.09, *p* = 0.020), but only four parents had scores consistent with moderate or severe symptoms. There was no significant change in the parents’ depression scores (47.30 ± 5.62) (Figure 2).

Parents’ anxiety and depression scores were significantly correlated both at baseline (r = 0.748, *p* < 0.001) and at the time of lockdown (r = 0.496, *p* = 0.001). 

Parents’ anxiety scores were significantly correlated with the parent-proxy anxiety scores both at baseline (r = 0.611, *p* < 0.001) and at the time of lockdown (r = 0.684, *p* < 0.001). Similarly, parents’ depression scores were significantly correlated with parent-proxy depression scores at both time points (baseline: r = 0.397, *p* = 0.009; lockdown: r = 0.387, *p* = 0.01). 

There were no significant differences between mothers and fathers in depression and anxiety scores at both time points. 

## 4. Discussion

In the current study, we investigated the emotional state and lifestyle habits of CCS and their parents during COVID-19-associated home confinement and compared them to levels documented prior to the pandemic. 

### 4.1. Lifestyle Habits

Increased unstructured time, the directive to stay indoors, and heightened stress associated with the COVID-19 pandemic led to widespread concerns about vulnerability to overeating, sedentary behavior, and weight gain [39]. However, data regarding lifestyle habits of children and adolescents during home confinement are inconsistent. A number of pediatric studies showed a higher consumption of nutritious food [17,18,19,20] and reduced consumption of fast food during the lockdown. Yet, the average intake of fried and sweet food increased [17,21]. In the current study, eating behaviors improved during home confinement—80% of the parents reported having more family meals and 56.4% reported ordering less take-out food compared to their usual routine. 

“Stay at home” orders during the COVID-19 outbreak limited opportunities for physical activity. Indeed, several studies reported a significant decrease in total physical activity in children and adolescents [18,21,22,23,24,25]. This finding is worrisome, as two studies showed a protective effect of exercise on the emotional status of children during the COVID-19 outbreak [9,12]. In the current study, the duration of physical activity decreased during the lockdown period, but not significantly, and the percentage of patients engaged in physical activity was unchanged. During the lockdown period, children engaged less in outdoor activities, but were able to find alternative indoor activities. This finding is consistent with the results of a previous study comprising adolescent and young adult (AYA) cancer survivors [40] and of a study comprising primary school adolescents from Poland [20], showing no significant change in exercise levels during the COVID-19 pandemic.

There is consistent evidence that sedentary/screen time is associated with unwanted weight gain in children [41], as well as with depression [42]. In the current study, leisure-related screen time nearly doubled during the lockdown period (947.3 ± 572.8 min/week vs. 477.0 ± 352.9 min/week, *p* < 0.001). Furthermore, six children (13.9%) spent time in front of screens only during the lockdown period. These results are consistent with results in a number of studies comprising healthy children and adolescents [20,21,22,23,24,25]. Notably, screen time in our cohort was considerably lower compared to the findings of a Canadian survey, where children spent 5.1 ± 3.5 h/day in front of screens during the COVID-19 outbreak, and adolescents 6.5 ± 3.3 h/day. Similar to our study, parents reported that leisure screen time was much higher than before the outbreak [25]. In a multinational study assessing the lifestyle habits of adolescents during the COVID-19 outbreak, watching TV during mealtimes was associated with poorer dietary quality [17]. In our cohort, 21.4% of the patients had their main meal while screen-watching more than three times a week during the lockdown period, similar to the baseline. Similar results were reported in a study comprising young adolescents from Poland [20].

### 4.2. Sleep

Multiple studies reported a shift in sleep time of children and adolescents during weekends and school holidays. The displaced bed and wake times may be associated with obesogenic behaviors such as screen-watching and snacking [41]. In the current study, the time of sleep was delayed by about two hours during the lockdown period, similar to the reported time of sleep during school holidays. Sleep duration increased significantly (*p* = 0.006) during the lockdown period. Similar findings were reported in several pediatric surveys [18,21,23,25] as well as in adults [14]. 

### 4.3. Depression and Anxiety

Several studies showed a significant association between a history of prior illness or chronic disease and adverse psychological impacts of the COVID-19 pandemic [1,6,26,43]. Our findings of a significant increase in anxiety scores in the parents’ self-reports as well as in their proxy-reports, can be explained by the parents’ perception of their children who recovered from cancer as more vulnerable than healthy children [27]. Similar to our results, a study comparing anxiety levels of 45 children diagnosed with cystic fibrosis (CF) and their mothers to a healthy control group during the COVID-19 outbreak found that mothers of children with CF had higher anxiety scores compared with mothers of healthy children. Interestingly, COVID-19 had no effect on the anxiety levels of children with CF [43]; a study comprising caregivers of CCS showed that 77% felt more anxious during the COVID-19 outbreak than before, while 64% of caregivers felt more depressed [44]. In another study, a moderate negative impact of COVID-19 on anxiety and mood was observed in caregivers as well as in adolescent survivors [40]. 

In our view, the physical vulnerability and the higher tendency of anxiety and depression among childhood cancer survivors and their parents [27,45] increase the susceptibility to anxiety, especially during times of uncertainty such as the lockdown period. The increase in anxiety rather than depression levels of parents, observed in our study, is compatible with findings from a survey conducted by the Israeli Central Bureau of Statistics (CBS) during the COVID-19 lockdown period, where 34% of 2792 individuals reported stress and anxiety compared to 16% who reported feelings of depression. In that survey, major concerns were being infected by SARS-CoV-2 and the economic ramifications of the lockdown, which could have resulted in anxiety [46]. Notably, most studies examining the emotional effects of the COVID-19 outbreak on both adults and children reported increased rates of depression as well as anxiety [1,2,3,4,5,6,7,8]. However, most of these studies did not include baseline data on anxiety and depression rates in the investigated populations and were based on convenience sampling.

### 4.4. Strengths and Limitations

The strengths of the current study consist of the availability of baseline data obtained within a short time period prior to the COVID-19 outbreak, and our well-defined cohort with a high response rate (77%) decreasing the possibility of selection bias. In addition, we analyzed a wide set of variables, comprehensively assessing lifestyle behaviors as well as the emotional status of parents and children. However, several limitations of our study should be considered. First, since other studies used different questionnaires to assess levels of anxiety and depression, we could not directly compare these rates in our participants to data from the general population in Israel or from other studies. Still, the availability of baseline data allowed us to directly assess the effect of the COVID-19 outbreak on mental health, which was not possible in most previous studies. In addition, the patients themselves filled out only the baseline questionnaires, so we could not directly assess changes in their emotional state during the lockdown period. Our decision to refrain from asking children to fill out questionnaires during the lockdown period was based on the fact that the baseline questionnaires were filled out by children without the presence of their parents, with the assistance of a research coordinator. As many of the participating children were relatively young and would not be able to answer the lockdown questionnaires online by themselves, it was felt that the parents’ assistance could skew the results. Importantly, at the baseline evaluation, there was no significant difference between the children’s self-report scores and the parents’ proxy report for both depression and anxiety, suggesting that the parents’ reports were a valid measure of their children’s emotional state. Finally, due to the online questionnaire being a self-report evaluation, the indicated levels of anxiety and depression may not be consistent with the evaluation of mental health professionals. However, a recent study comprising pediatric cancer patients showed good agreement between the child-reported and parent-reported PROMIS scores for depression and anxiety and an evaluation using a semi-structured face-to-face interview [32]. 

## 5. Conclusions 

In the current study, we have shown that childhood cancer survivors and their parents had higher levels of anxiety during the COVID-19-associated home confinement compared with the period before, similar to worldwide studies in healthy adults and children, as well as in those with chronic diseases [1,2,4,5,6,7,8,9,11,12,13,40,44]. Despite widespread concerns regarding lifestyle habits during the COVID-19 outbreak [14,16,17,18,21,22,23,24,25], in our cohort, eating behaviors improved, sleep duration increased, and physical activity remained unchanged during confinement, although screen exposure time increased significantly. 

Our findings suggest that at times of national crises, it is important to reach out to the vulnerable population of CCS and their caregivers, to identify families at risk of emotional distress, and to provide information and guidance by medical and psycho-social teams, pinpointing their past experiences and familial rituals as a resource. Given the positive association between high levels of physical activity, low levels of sedentary behaviors, and sufficient sleep with better mental health indicators among children and adolescents [42], we suggest assisting parents in establishing a daily healthy routine of activity and eating behavior. Although our results and recommendations refer to childhood cancer survivors and their parents, they may be generalizable to the healthy population as well, due to the common characteristics of lack of information, uncertainty, and increasing anxiety during future times of quarantine and isolation.

## Figures and Tables

**Figure 1 nutrients-15-03932-f001:**
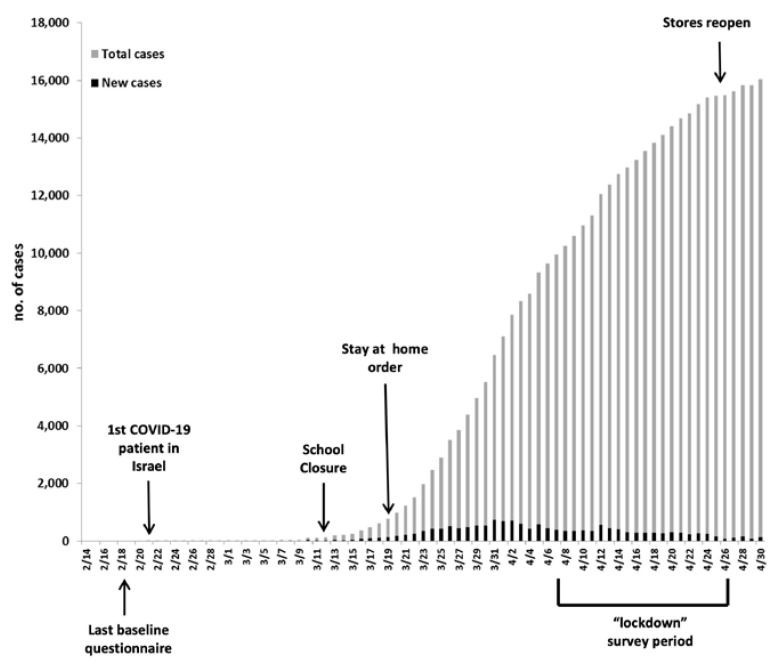
Time frame of the study in relation to COVID-19 in Israel. The last baseline questionnaire was administered on 18 February 2020, when the only COVID-19-related restrictions were self-quarantine orders for people arriving from Southeast Asia. Lockdown questionnaires were administered between 7 and 26 April 2020. The first COVID-19 patient was identified on 21 February. The number of cases increased during March, with a corresponding escalation in social distancing orders. School closure was announced on 12 March, and on 19 March, “stay at home” orders were issued. By 29 March, the rate of unemployment increased from 3.4% to 22.6%. Restrictions were gradually eased starting 19 April, and on 26 April, stores and many services and places of business were opened. Hence, parents who did not respond to the questionnaires by that time were not sent further reminders.

**Figure 2 nutrients-15-03932-f002:**
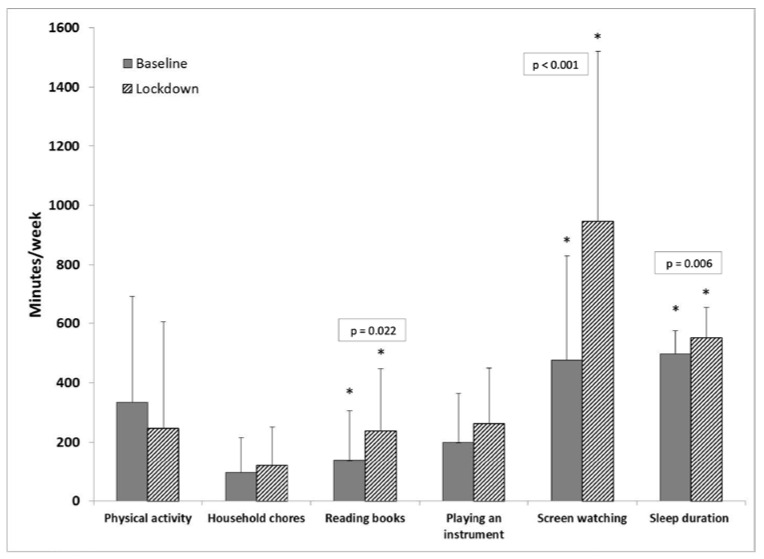
Physical and sedentary activities at baseline compared to lockdown. Screen watching, time devoted to reading books, and sleep duration increased significantly during the lockdown period. There was no significant change in time devoted to physical activity, playing instruments, or household chores.

**Figure 3 nutrients-15-03932-f003:**
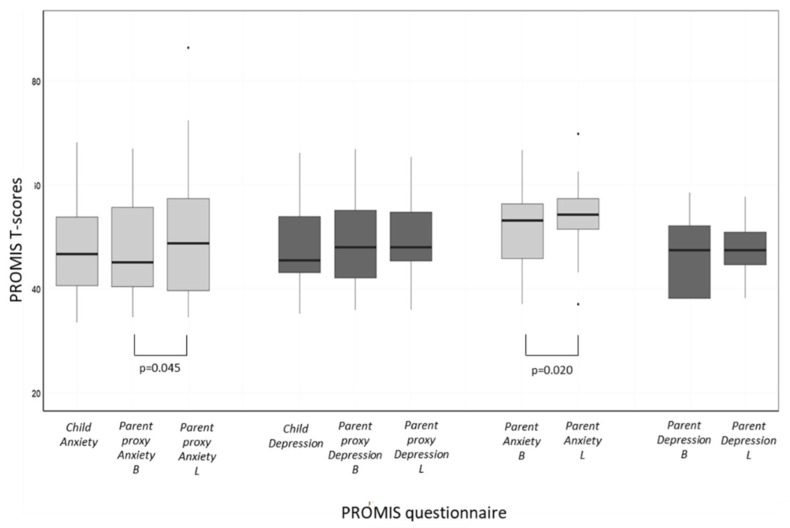
PROMIS depression and anxiety T-scores of children and parents at baseline (B) and during lockdown (L). Parents’ anxiety scores as well as parent-reported children’s anxiety scores increased significantly during the lockdown period. PROMIS = Patient-Reported Outcomes Measurement Information System.

**Table 1 nutrients-15-03932-t001:** Physical and sedentary activities at baseline vs. lockdown period.

Activity	Baseline		Lockdown		*p*-Value **
	% Performing	Minutes/Week *	% Performing	Minutes/Week *	
Total physical activity	90.7	334.7 ± 356.5	89.7	247.5 ± 359.1	0.115
Household chores	85.4	97.2 ± 118.2	85.7	121.9 ± 129.3	0.124
Reading books	56.1	138.9 ± 167.6	60.5	238.7 ± 208.8	0.022
Playing music	22	198.7 ± 166.1	19.0	263.3 ± 186.8	0.79
Screen time	78	477.0 ± 352.9	88.1	947.3 ± 572.8	<0.001

* Duration of activity, mean ± SD. ** Duration at baseline vs. lockdown, after log-transformation.

**Table 2 nutrients-15-03932-t002:** Types of physical activity at baseline and during the lockdown period.

	Baseline		Lockdown	
	n Performing	% Performing	n Performing	% Performing
Running	21	51.2	8	19.0
Swimming	6	14.6	1	2.4
Walking	23	57.5	16	39.0
Cycling	12	29.3	4	9.5
Gymnastics	13	31.7	7	16.7
Weight training, muscle strengthening	11	26.8	7	16.7
Ball games	17	41.5	13	31.0
Martial arts	1	2.4	2	4.8
Skateboarding/ rollerblading	2	4.9	-	-
Exercise while playing Wii, Xbox, Kinect, etc.	3	7.3	3	7.1
Dancing	11	26.8	9	20.9
Other *	8	19.5	11	26.8

* Push-ups, pull-ups, suspension training, calisthenics, treadmill walking or running.

## Data Availability

The data that support the findings of this study are available upon request from the senior author. The data are not publicly available because of privacy or ethical restrictions.

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
