# Peer review of "Impact of COVID-19 on Lifestyle Habits and Emotional State of Childhood Cancer Survivors and Their Parents"

_nutrients, 2023, doi:10.3390/nu15183932_

Round 1

Reviewer 1 Report

Dear Authors,

Your article is of interest but in the part about depression and anxiety e confuse and must be improved, our main objective in this revision is to help you understand where are the confusions, hoping that it will be a contribute for a better understanding for future lecturers.

Lines 111 to 113, please specify that the questionnaires were applied in their Hebraic version, translated and validated.

Line 116 “For the pediatric and parent-proxy PROMIS questionnaires” Did you applied a pediatric PROMIS questionnaire to adolescent and young adults childrens? This part assessment about depression and anxiety is confuse and need to be well clarified

Line 148 “mothers 31” please rectify. Or you inform only about the age of the parent or you give this information and add the medium age of the Mothers and the same for the fathers, because like it is why only the mothers?

Line 163: (p=1.0) please rectify to a more complete information (paired t test …)

Line 203 “average hour of sleep” probably it should be “average hour for going to bed” or something like that and not hour of sleep

Our main concern: Lines 216 to 221. You have to explain why for the children’s you have no data’s during the lockdown! With the data’s of the parents and the children’s baseline and during the lockdown, the study should have been more complete? Even for the discussion part on this topic you present data’s of other studies with children evolution during the lockdown but your study have not those information, only with the parents evolution and it turn your discussion and the conclusions of your discussion confuse. First, we suggest you to explain clearly why only the parents in B and L and children only in B. After don’t let mixture the concepts in the discussion and avoid proposing comparison and conclusion for children’s only on the basis of parents results.

Along the study you present children for subjects from 8,4 to 20,5 years, it is correct because there are the children’s of their parents, but in the other side children is not appropriate designation for adolescents and young adults. May be using the terminology sons should be better.

What about the BMI evolution of the sons subjects between B and L, maybe it should be associated with nutritional or lifestyle habits?

Discussion – Depression and anxiety. The studies used in this part of the discussion speak about anxiety and depression evolution in children with diseases during the covid period. But they have not comparison with your study where the evolution of those parameter in the CSC are not analysed. Only the parents, not the children’s. 

Line 317 to 323, in our opinion this study have very few to do with your study where the parent are concerned by the fragility of their CS child and express what they feel about their sons.

Line 345, can you affirm that “In the current study we have shown that childhood cancer survivors had higher levels of anxiety during the COVID-19 associated home confinement compared with the period before”? Can you also affirm that “In the current study we have shown that parents of childhood cancer survivors had higher levels of anxiety during the COVID-19 associated home confinement compared with the period before”? Witch questionnaires allow you to affirm that if you only have the PROMIS baseline of the sons and the parent-proxy PROMIS of the parents in B and L? Or we have not all the information in the methodology part or it is necessary to present your results and discussion in function of what you really have. In our interpretation to compare B to L you have only the opinion of the parents about their sons, this is not the personal anxiety or depression of the parents nor of their sons. Please expose that clearly along your article, include in your conclusions.

We wish you success in the reformulation of your article!

Author Response

Review # 1- Impact of COVID-19 on Lifestyle Habits and Emotional State of Childhood Cancer Survivors and Their Parents - Nutrient-2589428

We thank the reviewer for the thorough evaluation of our manuscript and for the constructive comments that helped to strengthen our manuscript.

Lines 111 to 113, please specify that the questionnaires were applied in their Hebraic version, translated and validated.

Reply: This has been added to the manuscript as suggested by the reviewer (lines 116-120).

Line 116 “For the pediatric and parent-proxy PROMIS questionnaires” Did you applied a pediatric PROMIS questionnaire to adolescent and young adults childrens? This part assessment about depression and anxiety is confuse and need to be well clarified

Reply: At the time of baseline evaluation the patients filled the PROMIS depression and anxiety questionnaires, while the parents filled the self-report PROMIS depression and anxiety questionnaires (i.e. their reflection on their own emotional state) as well as the parent-proxy questionnaires (i.e. their perception of their child's emotional state).  During the lockdown period, the parents filled the self-report depression and anxiety questionnaires and the parent-proxy questionnaires, while their children were not invited to answer questionnaires. This is now detailed in the METHODS section (line 120-122).

Notably, the definition of "pediatrics" currently includes infants, children, and adolescents up to the age of 21 years. Thus, all patients included in our study were at the pediatric age group.

Line 148 “mothers 31” please rectify. Or you inform only about the age of the parent or you give this information and add the medium age of the Mothers and the same for the fathers, because like it is why only the mothers?

Reply: There were 31 participating mothers and 12 fathers. The mean age presented is of all 43 participating parents. The number of participating fathers was added to the manuscript (lines 154).

Line 163: (p=1.0) please rectify to a more complete information (paired t test …)

Reply: The p-value of 1.0 was derived from a comparison using the McNemar Test for paired data. This is detailed in the statistical analysis section (line 146).

Line 203 “average hour of sleep” probably it should be “average hour for going to bed” or something like that and not hour of sleep

Reply: This sentence was changed according to the reviewer's suggestion (line 208).

Our main concern: Lines 216 to 221. You have to explain why for the children’s you have no data’s during the lockdown! With the data’s of the parents and the children’s baseline and during the lockdown, the study should have been more complete? Even for the discussion part on this topic you present data’s of other studies with children evolution during the lockdown but your study have not those information, only with the parents evolution and it turn your discussion and the conclusions of your discussion confuse. First, we suggest you to explain clearly why only the parents in B and L and children only in B. After don’t let mixture the concepts in the discussion and avoid proposing comparison and conclusion for children’s only on the basis of parents results.

Reply: As detailed in the revised METHODS section, at the time of baseline evaluation the patients filled the pediatric PROMIS depression and anxiety questionnaires as well as the lifestyle habits questionnaire, while the parents filled the self-report PROMIS depression and anxiety questionnaires as well as the parent-proxy questionnaires. During the lockdown period, we asked the parents to answer the adult self-report depression and anxiety questionnaires and the parent-proxy questionnaires (as they did at the baseline assessment), as well as a shortened version of the lifestyle habits questionnaires. This is now clarified in the METHODS section (lines 118-122 + lines 129-130 of MABAT).

We agree with the reviewer that having the children fill the questionnaire would strengthen our findings, however our decision to refrain from asking children to fill questionnaires during the lockdown period was based on the fact that the baseline questionnaires were filled by children without the presence of their parents, with the assistance of our research coordinator. As many of the participating children were relatively young, they would not be able to answer the lockdown questionnaires online by themselves, and we felt that parents’ assistance could skew the results (i.e. that the parents may influence the child’s response to the questionnaire).  This explanation was added to the manuscript (lines 345-347).

In the discussion we state that we found "a significant increase in anxiety scores in the parents' self-reports as well as in their proxy-reports" that "can be explained by the parents' perception of their children who recovered from cancer as more vulnerable than healthy children" (line 306), thus acknowledging the fact that we did not directly assess the emotional state of the children during the lockdown period. Later in the discussion we compare our findings to those of other studies assessing caregivers of children with chronic disease and of childhood cancer survivors (lines 311-316).

In accordance with the reviewer's comment, and to further clarify this issue we made the following changes in the discussion:

  1. We deleted the reference to patients' own experiences (reference 26).
  2. We added in line 320 the word "parents".
  3. Added the lack of patients'/children's lockdown self-report questionnaires as a limitation of our study (lines 340-350).

Importantly, at the baseline evaluation there was no significant difference between the children's self-report scores and the parents' proxy report for both depression and anxiety (lines 227-229), suggesting that the parents' reports were a valid measure of their children's emotional state. Regarding the lifestyle habits questionnaires, since during the lockdown period parents were with their children at all times, we believe they could reliably report activity levels and eating habits of their children,  

Along the study you present children for subjects from 8,4 to 20,5 years, it is correct because there are the children’s of their parents, but in the other side children is not appropriate designation for adolescents and young adults. May be using the terminology sons should be better.

Reply: We thank the reviewer for this comment and changed to "patients" or "survivors" where appropriate.

What about the BMI evolution of the sons subjects between B and L, maybe it should be associated with nutritional or lifestyle habits?

Reply: Unfortunately, during the lockdown patients did not attend the hospital, so these data are not available. 

Discussion – Depression and anxiety. The studies used in this part of the discussion speak about anxiety and depression evolution in children with diseases during the covid period. But they have not comparison with your study where the evolution of those parameter in the CSC are not analysed. Only the parents, not the children’s. 

Reply: Indeed, our study mainly shows changes in the parents' emotional state and in their perception of their children's emotional state. Accordingly, in the DISCUSION we did not directly comment on the emotional state of our patients but rather wrote that anxiety scores in the parents' self-reports as well as in their proxy-reports, can be explained by the parents' perception of their children who recovered from cancer as more vulnerable than healthy children" (lines 306). Furthermore, we cite and refer to other studies assessing caregivers of children with chronic disease (ref 43 in the revised manuscript) and of childhood cancer survivors (references 40 & 44). Importantly, at the baseline evaluation there was no significant difference between the children's self-report scores and the parents' proxy report for both depression and anxiety (lines 347-350), suggesting that the parents' reports were a valid measure of their children's emotional state.

As described above, in light of the reviewer's comment, and in order to clarify this issue we made the following changes in the discussion:

  1. We deleted the reference to patients' own experiences (reference 26).
  2. We added in line 320 the word "parents".
  3. Added the lack of patients'/children's lockdown self-report questionnaires as a limitation of our study (lines 340-350).

Line 317 to 323, in our opinion this study have very few to do with your study where the parent are concerned by the fragility of their CS child and express what they feel about their sons.

Reply: The study in question (reference 43) showed that mothers of children with cystic fibrosis had higher anxiety levels during the COVID-19 outbreak compared to mothers of healthy children. In our opinion this is relevant to the current study, as we believe that parents of childhood cancer survivors perceived their children as more vulnerable (and thus more susceptible to COVID-19 infection) than healthy children, similar to parents with chronic diseases such as CF. Hence, we prefer to keep this reference in the discussion, and leave this issue to the discretion of the reviewer and the editor.

Line 345, can you affirm that “In the current study we have shown that childhood cancer survivors had higher levels of anxiety during the COVID-19 associated home confinement compared with the period before”? Can you also affirm that “In the current study we have shown that parents of childhood cancer survivors had higher levels of anxiety during the COVID-19 associated home confinement compared with the period before”? Witch questionnaires allow you to affirm that if you only have the PROMIS baseline of the sons and the parent-proxy PROMIS of the parents in B and L? Or we have not all the information in the methodology part or it is necessary to present your results and discussion in function of what you really have. In our interpretation to compare B to L you have only the opinion of the parents about their sons, this is not the personal anxiety or depression of the parents nor of their sons. Please expose that clearly along your article, include in your conclusions.

Reply: As detailed above, as well as in the revised METHODS section (lines 118-122), at the time of baseline evaluation the patients filled the PROMIS Pediatric depression and anxiety questionnaires, while the parents filled the self-report PROMIS depression and anxiety questionnaires as well as the parent-proxy questionnaires. Thus, we had baseline data for the children self-report questionnaires, the parents' self-report questionnaires and for the proxy-reports. Lockdown data included the parents' self-report and the parent-proxy report. This is also visualized in figure 3.

Accordingly, we could comment on the changes in the parents' self-report scores of depression and anxiety questionnaires as well as on changes in their perception of the emotional state of their children, and indeed in the caption of figure 3 we wrote: "Parents' anxiety scores as well as parent-reported children's anxiety scores increased significantly during the lockdown period".

As noted above, we revised the discussion to clarify this issue (line 320) and added the lack of patients' lockdown questionnaires as a limitation (lines 340-350).

We wish you success in the reformulation of your article!

Reply: we thank the reviewer again for the constructive comments and hope you will found our manuscript suitable for publication in NUTRIENTS.

Reviewer 2 Report

Thank you for inviting me to review the manuscript titled Impact of COVID-19 on Lifestyle Habits and Emotional State of Childhood Cancer Survivors and Their Parents. The topic of this paper is interesting and important. The study investigates the change in lifestyle habits and mental health in the population of childhood cancer survivors and their parents due to the COVID-19 pandemic. The manuscript can be improved. I have included the list of my recommendations below.

In the abstract, the names of questionnaires should be included.

The introduction is well-structured and presents the background based on listing consequences of isolation presented in the literature, including changes in mental health, quality of consumed food and the amount of physical activity.

Line 79. Can you please add a reference to “larger-scale study investigating chronic stress and lifestyle habits in CCS and their parents”? If not available, please add some details such as the place of conduct the way of data collection, etc.

Line 90. What do you mean by the term “non-participants”? Please clarify. If such people were excluded due to eligibility criteria, please formulate that in a clear manner.

In the methods, it is not clearly stated if the same people were assessed during the baseline and lockdown survey. Please clarify.

Lines 119-125, please correct font. Also, references should be added to the reference list and not cited in the text.

Line 122 – Please add references for the following statement: The questionnaire has been used in a national survey and validated in healthy Israeli children and adolescents.

Line 154 – Please explain the abbreviation – SDS.

Table 1. This comparison is not clear. What is compared? Percentages or duration? And with what tests? The two comparisons could result in different levels of statistical significance.

Line 186 – How was the screen time calculated? Was online school time included in this? Or was this just screen time associated with video games? If education was included this is not very informative, as those 6 children could use computers only for educational/school purposes.

Line 209 – There is no title for Figure 2. Please add a title.

Line 245 – please add a title to Figure 3.

I would recommend adding some additional thought to the discussion/limitations. First, screen exposure is not clear in this study, so the reader does not know if this is for leisure or educational purposes. Children would spend some time sitting on the school bench similar to time spent at their computers anyway. My second doubt is the overall level of physical activity. If this level was not changed during lockdown, I would assume that it was very low at baseline – probably far too low for such a population that should have a healthy lifestyle. And finally, I feel that you make an assumption that food ordered from vendors and not cooked at home is unhealthy. I do not agree with this as people can buy healthy meals as take-away food as well.

The manuscript should be more tidy. There are some typos, capitalisation errors, and hyphenation errors.

Author Response

 Review # 2- Impact of COVID-19 on Lifestyle Habits and Emotional State of Childhood Cancer Survivors and Their Parents - Nutrient-2589428

Thank you for inviting me to review the manuscript titled Impact of COVID-19 on Lifestyle Habits and Emotional State of Childhood Cancer Survivors and Their Parents. The topic of this paper is interesting and important. The study investigates the change in lifestyle habits and mental health in the population of childhood cancer survivors and their parents due to the COVID-19 pandemic.

We thank the reviewer for the comprehensive assessment of our manuscript and for the helpful comments and suggestions.

In the abstract, the names of questionnaires should be included.

Reply: This information was added to the abstract according to the reviewer's suggestion.

The introduction is well-structured and presents the background based on listing consequences of isolation presented in the literature, including changes in mental health, quality of consumed food and the amount of physical activity.

We thank the reviewer for this comment.

Line 79. Can you please add a reference to “larger-scale study investigating chronic stress and lifestyle habits in CCS and their parents”? If not available, please add some details such as the place of conduct the way of data collection, etc.

Reply: The larger/"parent" study has been conducted in our clinic and its results are not yet published. As described in the METHODS section (lines 84-93), patients' recruitment started on July 2019 and 56 parent-child dyads completed the study questionnaires by February 18th (before the first COVID-19 patient was detected in Israel).  During the lockdown period, we approached 54 of these parents (two mothers were excluded as explained below), and 43 of them answered questionnaires for the current study. Once COVID-19 social distancing restrictions were removed we continued to recruit participants to the original study (results are now being analyzed).

Line 90. What do you mean by the term “non-participants”? Please clarify. If such people were excluded due to eligibility criteria, please formulate that in a clear manner.

Reply: As described above, by the time Stay-At-Home orders were issued, 56 parent-child dyads completed the baseline study questionnaires. Of these, 54 parents were approached and offered participation in the current study (one mother was excluded as her daughter had relapse of her disease shortly after responding to the baseline questionnaire, and another mother was excluded due to her own medical circumstances). Of the 54 parents who were approached, 43 eventually answered the lockdown questionnaires. The 11 non-participants included two who did not respond to recruitment phone calls, and nine parents that agreed to participate but did not fill the questionnaire. The relevant paragraph in the METHODS was edited to clarify this (lines 94-96).

In the methods, it is not clearly stated if the same people were assessed during the baseline and lockdown survey. Please clarify.

Reply: Indeed, as described above, the same people were assessed during the baseline and lockdown survey. This is now clarified in the METHODS section (lines 118-122).

Lines 119-125, please correct font. Also, references should be added to the reference list and not cited in the text.

Reply: This has been done according to the reviewer's suggestion.

Line 122 – Please add references for the following statement: The questionnaire has been used in a national survey and validated in healthy Israeli children and adolescents.

Reply: Reference 35 in the revised manuscript contains the methodology of the national survey as well as its results.

Line 154 – Please explain the abbreviation – SDS.

Reply:  In accordance with the reviewer's comment, "standard deviation score" was added to the manuscript (line 160).

Table 1. This comparison is not clear. What is compared? Percentages or duration? And with what tests? The two comparisons could result in different levels of statistical significance.

Reply:  Table 1 presents comparisons between duration (in minutes per week) of total physical activity and of different types of activity at the time of baseline assessment and during the lockdown. This is now depicted as a footnote to the table (see below). As detailed in the "statistical methods" section, the comparison was done using the paired t-test (line 142).

Activity

Baseline

Lockdown

p-value**

% performing

minutes/week*

% performing

minutes/week*

Total physical activity

90.7

334.7±356.5

89.7

247.5±359.1

0.115

Household chores

85.4

97.2±118.2

85.7

121.9±129.3

0.124

Reading books

56.1

138.9±167.6

60.5

238.7±208.8

0.022

Playing music

22

198.7±166.1

19.0

263.3±186.8

0.79

Screen time

78

477.0±352.9

88.1

947.3±572.8

<0.001

*Duration of activity, mean±SD  ** Duration at baseline vs. lockdown, after log-transformation.

 Line 186 – How was the screen time calculated? Was online school time included in this? Or was this just screen time associated with video games? If education was included this is not very informative, as those 6 children could use computers only for educational/school purposes.

Reply: The relevant question in the lifestyle questionnaire (reference 35) reads: "Did you watch television/video shows or computer programs in the past week? (not in order to prepare homework)", and then asks "how many times a week" and "what is the duration of the activity each time".

Thus, our findings regarding screen time use relate specifically to leisure related screen time. This is now clarified in the Abstract, in the RESULTS section (line 190) and in the DISCUSSION (line 281).

Line 209 – There is no title for Figure 2. Please add a title.

Reply: The title "Physical and sedentary activity, baseline compared to lockdown" has been added.

Line 245 – please add a title to Figure 3.

Reply: The title " PROMIS depression and anxiety T-scores of children and parents at baseline and during lockdown" has been added.

Screen exposure is not clear in this study, so the reader does not know if this is for leisure or educational purposes. Children would spend some time sitting on the school bench similar to time spent at their computers anyway.

Reply: As discussed above, the relevant question in the lifestyle questionnaire (reference 35) referred specifically to leisure-related screen time. This is now clarified in the Abstract, in the RESULTS section (line 190), and in the DISCUSSION (line 281).

My second doubt is the overall level of physical activity. If this level was not changed during lockdown, I would assume that it was very low at baseline – probably far too low for such a population that should have a healthy lifestyle.

Reply: As presented in the manuscript (lines 183-188 and Table 1), on the baseline questionnaires 90.7% of the patients reported engaging in physical activity, with a mean of 334.7 minutes/week i.e. 47.8 minutes per day. This is somewhat lower, but not much lower, than the current pediatric recommendations of at least 60 minutes of moderate or vigorous intensity physical activity a day. During the lockdown period, there was no change in the percentage of patients engaging in physical activity (89.7%), while the duration of activity was lower (247.5±359.1 minutes/week), but not significantly so. Children adjusted the type of activity to the lockdown conditions, engaging less in outdoor activities such as running, swimming, walking, and cycling, and shifting to indoor activities such as weight training, muscle strengthening, martial arts and dancing (see Table 2). As described in the manuscript, similar results were reported by other investigators (references 20 & 40, lines 276-278, 293-294).

And finally, I feel that you make an assumption that food ordered from vendors and not cooked at home is unhealthy. I do not agree with this as people can buy healthy meals as take-away food as well.

Reply: We respectfully disagree with the reviewer on this point. Our personal experience with ordering takeaway food in Israel, as well as our experience from informal conversations with patients, suggest that takeaway food tends to be mainly fast food. This observation is supported by a recent study from the United Kingdom that showed that during the pandemic takeaway/delivery services were seldom used to access full-service retailers, but the use of delivery services to access fast food was high [1]. Furthermore, the 2023 guidelines for the evaluation and treatment of Children and adolescents with obesity of the American Academy of Pediatrics state that: "Take-away food has also been associated with high BMI. Hence, eating outside of the home—irrespective of the type of restaurant establishment visited—is associated with higher risk of weight or BMI gain" [2].

  1. Fong M, Scott S, Albani V, Brown H. The Impact of COVID-19 Restrictions and Changes to Takeaway Regulations in England on Consumers' Intake and Methods of Accessing Out-of-Home Foods: A Longitudinal, Mixed-Methods Study. Nutrients. 2023 Aug 18;15:3636.
  2. Hampl SE, Hassink SG, Skinner AC, Armstrong SC, Barlow SE, Bolling CF, Avila Edwards KC, Eneli I, Hamre R, Joseph MM, Lunsford D, Mendonca E, Michalsky MP, Mirza N, Ochoa ER, Sharifi M, Staiano AE, Weedn AE, Flinn SK, Lindros J, Okechukwu K. Clinical Practice Guideline for the Evaluation and Treatment of Children and Adolescents With Obesity. Pediatrics. 2023 Feb 1;151(2):e2022060640. doi: 10.1542/peds.2022-060640.

The manuscript should be more tidy. There are some typos, capitalisation errors, and hyphenation errors.

Reply: In light of the reviewer's comment, the manuscript has been carefully reviewed by a native English speaking editor and errors have been corrected.

Reviewer 3 Report

Dear Authors,

I am very grateful that I had the opportunity to make this review. The COVID-19 pandemic was a difficult and stressful period for all the people.

Here you find my comments:

Materials and methods

Line 124 Please describe the questionnaire, the items that you investigated

Results

The results must be reorganized, there are unclear

Line 149 children males=24 and females?

Line 150 It is unclear the sentence, please rewrite

Line 148 mothers=31. It is unclear , there are 31 mothers?

Line 154 SDS? Please explain..abbreviation of..what?

Line 167 80% and the number?

Line 182. It is unclear the sentence

Line 187 Six children…the percentage? Please express always the results as  number and percentage

Line 190 You stated Table 1 and figure 2. There has to be two different explanations. There is unclear. Table 1 express the same results as figure 2? It is not correct

Fig.2  The numbering and the title of a figure is below the figure

Line 206

Sleep….Where is the figure?

Line 245 Figure 3..? below the figure

References

Must be written according to the MDPI  recommendation

Kind regards

Author Response

Review # 3 - Impact of COVID-19 on Lifestyle Habits and Emotional State of Childhood Cancer Survivors and Their Parents - Nutrient-2589428

Dear Authors, I am very grateful that I had the opportunity to make this review. The COVID-19 pandemic was a difficult and stressful period for all the people.

We thank the reviewer for taking the time to review our manuscript and for the valuable comments and suggestions.

Materials and methods

Line 124 Please describe the questionnaire, the items that you investigated

Reply: The English version of the complete questionnaire can be viewed online – a link is now provided as reference 35. 

Results

The results must be reorganized, there are unclear

Reply: The RESULTS section has been revised according to the comments of the three reviewers.

Line 149 children males=24 and females?

Reply: There were 19 female participants. This information was added to the manuscript (line 155).

Line 150 It is unclear the sentence, please rewrite

Reply: This sentence was changed as suggested by the reviewer (line 156 in the revised manuscript).

Line 148 mothers=31. It is unclear , there are 31 mothers?

Reply: There were 31 participating mothers and 12 fathers. The number of participating fathers was added to the manuscript (line 154).

Line 154 SDS? Please explain..abbreviation of..what?

Reply:  In accordance with the reviewer's comment, "standard deviation score" was added to the manuscript (line 160).

Line 167 80% and the number?

Reply: The number (n=34) has been added (line 172)

Line 182. It is unclear the sentence

Reply: Sorry, but we are not sure what sentence the reviewer refers to (the line numbers seem to have somewhat shifted).

Line 187 Six children…the percentage? Please express always the results as number and percentage

Reply: The percentage (14%) has been added (line 192).

Line 190 You stated Table 1 and figure 2. There has to be two different explanations. There is unclear. Table 1 express the same results as figure 2? It is not correct

Reply: Table 1 depicts the percentage of patients performing physical and sedentary activities, and time devoted to each activity in minutes/week, comparing findings at baseline to the lockdown period. Figure 2 presents the time devoted to the various activities at baseline vs. the lockdown period in a graphic fashion. So, indeed there is some overlap between the table and the figure. We added a title to Figure 2, which may help to clarify this issue. 

Fig.2  The numbering and the title of a figure is below the figure

Reply: We added a title to Figure 2.

Line 206: Sleep….Where is the figure?

Reply: The two last (right-sided) columns of Figure 2 depict sleep duration at the time of baseline and during the lockdown period.

Line 245 Figure 3..? below the figure

Reply: We added a title to Figure 3.

References: Must be written according to the MDPI recommendation

Reply: This has been done.

Kind regards

We thank the reviewer again for the constructive comments and hope you will find our manuscript suitable for publication in NUTRIENTS.

Round 2

Reviewer 1 Report

Thank you for taking in count all our observations, for future readers, we are sure it will be easier to understand. You did well and we hope you success in this publication.

Best regards

Reviewer 2 Report

Thank you for your revision. I do not have additional comments.

Reviewer 3 Report

Dear Authors,

Thank you for the opportunity to review your paper.

Please write the number of figures and the title below the figure.

Kind regards